# Higher Lead and Lower Calcium Levels Are Associated with Increased Risk of Mortality in Malaysian Older Population: Findings from the LRGS-TUA Longitudinal Study

**DOI:** 10.3390/ijerph19126955

**Published:** 2022-06-07

**Authors:** Theng Choon Ooi, Devinder Kaur Ajit Singh, Suzana Shahar, Razinah Sharif, Nurul Fatin Malek Rivan, Asheila Meramat, Nor Fadilah Rajab

**Affiliations:** 1Centre for Healthy Ageing and Wellness, Faculty of Health Sciences, Universiti Kebangsaan Malaysia, Jalan Raja Muda Abdul Aziz, Kuala Lumpur 50300, Malaysia; ooithengchoon@ukm.edu.my (T.C.O.); devinder@ukm.edu.my (D.K.A.S.); suzana.shahar@ukm.edu.my (S.S.); razinah@ukm.edu.my (R.S.); fatinmalek93@gmail.com (N.F.M.R.); 2Faculty of Health Sciences, Gong Badak Campus, Universiti Sultan Zainal Abidin, Kuala Nerus 21300, Malaysia; asheilameramat@unisza.edu.my

**Keywords:** calcium, lead, mortality, older adults, trace elements

## Abstract

The main objective of this study is to determine the association of various trace elements’ status with the 5-year mortality rate among community-dwelling older adults in Malaysia. This study was part of the Long-term Research Grant Scheme—Towards Useful Ageing (LRGS-TUA). The participants were followed up for five years, and their mortality status was identified through the Mortality Data Matching Service provided by the National Registration Department, Malaysia. Of the 303 participants included in this study, 34 (11.2%) participants had died within five years after baseline data collection. As compared to the survivors, participants who died earlier were more likely (*p* < 0.05) to be men, smokers, have a lower intake of total dietary fiber and molybdenum, higher intake of manganese, lower zinc levels in toenail samples, lower calcium and higher lead levels in hair samples during baseline. Following the multivariate Cox proportional hazard analysis, lower total dietary fiber intake (HR: 0.681; 0.532–0.871), lower calcium (HR: 0.999; 95% CI: 0.999–1.000) and higher lead (HR: 1.309; 95% CI: 1.061–1.616) levels in hair samples appeared as the predictors of mortality. In conclusion, higher lead and lower calcium levels are associated with higher risk of mortality among community-dwelling older adults in Malaysia. Our current findings provide a better understanding of how the trace elements’ status may affect older populations’ well-being and mortality rate.

## 1. Introduction

Global shifting toward aging populations is one of the main issues in the 21st century [1]. The proportion of the older population is increasing mainly due to the reduction in fertility rate and increase in life expectancy, and this phenomenon is more evident in developing compared to developed countries. The improvement in life expectancy is partly due to reducing mortality against infectious diseases. Meanwhile, non-communicable diseases and geriatric syndromes (e.g., cognitive decline and physical frailty) have become the major health threats among the older population, probably due to lifestyle changes, including diet [2]. Consequently, there are higher numbers of older people experiencing illnesses, disabilities, and dependency, posting a significant burden on global health. Preventive medicine and reducing impairments and disabilities due to diseases are the leading solutions to reduce social and health burdens [3].

The predictors and risk factors associated with mortality in older populations have been determined in many studies. Age and sexes are well-known unmodifiable risk factors for mortality among older populations. Advancing age is usually accompanied by physiological deterioration and additional exposure to the other risk factors, which eventually increase the possibility of death with aging [4,5]. In addition, the mortality rate is higher in males than females due to biological, behavioral and lifestyle differences [6,7]. Human longevity could also be partly attributed to certain inherited genetic factors and the interaction of these genes with environmental factors in their later life [8]. Meanwhile, modifiable risk factors such as psychosocial (depression, social isolation, etc.) and functional status (limited capacity to perform activities of daily living (ADL) and instrumental activities of daily living (IADL)), socioeconomic level (financial difficulties, education level), lifestyle (smoking, alcohol consumption, physical inactivity, etc.) and dietary habits (malnutrition and obesity), cognitive impairment and comorbidities/multimorbidity are associated with mortality in older adults [9,10,11,12,13,14]. Identifying such risk factors may help the policymakers plan and develop strategies and preventive measures to improve later-life health and reduce the risk of disabilities and dependency among the older population. This may eventually help improve the well-being and quality of life of older people while reducing the socioeconomic burden of the nation.

Although several risk factors associated with mortality have been identified, there is limited study focusing on the relationship between the trace element status and mortality rate in older people. The trace elements’ status in the body, accumulated either via environmental exposure or dietary intake, may affect the health condition and well-being and, eventually, the mortality rate of individuals [15,16,17]. This is because trace elements such as calcium, zinc, iron and selenium are essential trace elements that play an essential role in maintaining the normal physiological function of the body [18]. On the other hand, lead, aluminum and cadmium have no known physiological function but can cause toxicity once accumulated [18]. Moreover, the lack of certain essential trace elements and accumulation of toxic elements can cause oxidative stress and DNA damage in the cells, which may implicate the pathogenesis and progression of various diseases [19,20,21]. Previously, our in-house data indicated that advancing age, male, unmarried, smoking, higher fasting blood glucose level, lower serum albumin level, scoring poorer in the Timed Up and Go (TUG) test and lower intake of total dietary fiber were the predictors of mortality among community-dwelling older adults in Malaysia. However, the impact of various essential and toxic trace elements, as well as the oxidative stress and DNA damage biomarkers on mortality remain unclear. Hence, this study aims to determine the association of various trace elements status, oxidative stress and DNA damage biomarkers with the 5-year mortality rate among community-dwelling older adults in Malaysia. Results from this study may help fill the knowledge gap and provide a better understanding of the impact of trace elements on the well-being and life expectancy of older populations.

## 2. Materials and Methods

### 2.1. Participants

This study was part of the Long-term Research Grant Scheme—Towards Useful Ageing (LRGS-TUA), approved by the Medical Research and Ethics Committee of Universiti Kebangsaan Malaysia (UKM 1.5.3.5/244/NN-060-2013) and was conducted in accordance with the Declaration of Helsinki. Older adults aged 60 years and above from different regions of Peninsular Malaysia were recruited through a stratified random sampling method during baseline data collection. Details regarding the methodology on sampling methods, inclusion and exclusion criteria for the study were described in detail previously [22,23]. Written informed consent was obtained from each participant before participating in this study. Out of the total 2322 participants recruited during baseline data collection, only 303 randomly selected participants for an in-depth trace elements and biomarkers analysis were included in this study.

### 2.2. Data Collection

During baseline data collection, participants were interviewed face-to-face by trained enumerators, and information on sociodemographic data and medical history was collected using a standardized questionnaire, as described previously [22]. Participants’ body height and weight were measured using a portable SECA 206 portable body meter (Seca, Hamburg, Germany) and Tanita digital lithium weighing scale (Tanita, Tokyo, Japan), respectively. Then, the body mass index (BMI) of participants was calculated by using the formula “body weight (kg)/height (m)^2^”. A validated Dietary Habits Questionnaire was used to record the usual dietary intake of each participant in a week [24]. The depression status of the participants was determined using the Malay version Geriatric Depression Scale-15. Then, the participants were followed up for five years, and their mortality status was identified through the Mortality Data Matching Service provided by the National Registration Department (NRD), Malaysia.

### 2.3. Blood Samples Collection

The fasted venous blood was collected in the blood collection tubes by a trained phlebotomist. The heparinized whole blood samples were used to analyze DNA damage, as detected using the Alkaline Comet Assay. Meanwhile, the plasma samples separated from the heparinized tube were used to determine the superoxide dismutase (SOD) activities and malondialdehyde (MDA) levels [25].

### 2.4. Alkaline Comet Assay

Alkaline Comet Assay was performed as described previously by Meramat et al. [25]. First, 40 μL of whole blood were mixed thoroughly with 0.6% low melting point agarose (Sigma St. Louis, MO, USA) and laid on hardened 0.6% normal melting agarose (Sigma, St. Louis, MO, USA). The agarose was allowed to solidify and subsequently placed in a chilled lysis buffer (2.5 M NaCl, 100 mM EDTA, 10 mM Tris, and 1% Triton-X) for lysis to occur. Slides were then incubated in an electrophoresis buffer (0.3 N NaOH, 1 mM EDTA) for 20 min to facilitate DNA unwinding. Electrophoresis was performed under 25 V, 300 mA for 20 min. Subsequently, slides were rinsed with neutralizing buffer (400 mM Tris) thrice before staining with 10 µg/mL ethidium bromide solution (Sigma, St. Louis, MO, USA). Slides were kept overnight before being observed under a Nikon Eclipse TS-100 fluorescence microscope (Nikon, Tokyo, Japan) and analyzed with Comet Assay IV software (Instem-Perceptive Instruments Ltd., Suffolk, Halstead, UK).

### 2.5. Determination of SOD Activities and MDA Levels

The SOD activities and MDA levels were performed as previously described [25]. For the determination of SOD activities, 20 µL of plasma sample was mixed with 1 mL of substrate reagent (27 mL of 50 mM potassium phosphate buffer containing 0.1 mM EDTA at pH 7.8 mixed with 1.5 mL of L-methionine solution (30 mg/mL), 1 mL of nitro blue tetrazolium (NBT).2HCl solution (1.41 mg/mL) and 0.75 mL of 1% Triton-X 100 solution) before the addition of 10 µL of riboflavin solution (0.044 mg/mL) in dark condition. The mixture solution was incubated in an aluminum box under the illumination of two Sylvania GroLux fluorescent lamps (18 watts) to allow the photochemical reaction to occur. After 7 min of incubation, the absorbance of the mixture solution was determined using a microplate reader at the wavelength of 560 nm.

On the other hand, to determine the MDA levels, 100 µL of plasma sample was diluted with 400 µL of distilled water first before mixing with 2.5 mL of 20% trichloroacetic acid (TCA) solution (*w/v*). The mixture solution was allowed to be incubated at room temperature for 15 min before adding 1.5 mL 0.67% thiobarbituric acid (TBA) solution (*w/v*). The mixture was then heated in a water bath at 100 °C for 30 min and allowed to cool down to room temperature after heating. Then, 4 mL of n-butanol solution was added to the mixture and mixed thoroughly using a vortex for 3 min. After centrifuging for 10 min at 2000× *g*, the supernatant was transferred to a cuvette. The absorbance of the supernatant was quantified using a spectrophotometer at the wavelength of 532 nm, and the MDA levels of each sample were determined by using 1,1,3,3-tetraethoxypropane (TEP) solution as the standard reference.

### 2.6. Trace Element Analysis

A trained enumerator collected the toenail and hair samples from each participant. The toenail samples were clipped from all toes of participants by using a stainless-steel nail clipper, while the hair samples were plucked from different regions of the head with stainless steel tweezers. The collected toenail and hair samples were stored separately using sealed plastic bags at room temperature. Then, trace element analysis was performed as previously described [26]. Generally, both toenail and hair samples were subjected to a series of washing and digestion steps prior to trace element analysis. In general, 10 mg of the toenail or hair samples were washed in sequence using acetone (R&M, London, UK), deionized water, and 0.5% Triton X (Sigma, St. Louis, MO, USA). The samples were stirred while being washed and were rinsed with deionized water at the end of the washing step to remove the remaining reagents. After rinsing, the samples were put inside an oven overnight and dried at 60 °C. Next, samples were digested with 2 mL of 65% nitric acid (R&M, London, UK) at 250 °C. After 15 min of incubation, 0.2 mL of 30% hydrogen peroxide (Merck KGaA, Darmstadt, Germany) was added and topped up to the final volume of 10 mL using deionized water. The digested samples’ trace elements were analyzed using Inductively Coupled Plasma-Mass Spectrometry (Elan 9000, PerkinElmer, Shelton, CT, USA). All reagents used for the trace elements analysis were of analytical grade.

### 2.7. Statistical Analysis

The data were analyzed using the Statistical Package for the Social Sciences version 25.0 (IBM Corp, Armonk, New York, NY, USA). Descriptive analysis was used to study the sociodemographic data, fall history, anthropometric, medical history, biomarkers data, and participants’ trace element status. An independent T-test and Chi-square test were used for univariate comparison between the survival and death groups. Then, all the significant variables (*p* < 0.05) in the univariate tests were included in the multivariate Cox proportional hazard analysis and were adjusted with other well-known confounding factors of mortality (age, sex, ethnicity, smoking habits and multimorbidity status). The significant variables (*p* < 0.05) in the final model were considered to be associated with the incidence of mortality among community-dwelling older adults.

## 3. Results

Our current findings showed that 34 (11.2%) participants had passed away within five years after baseline data collection. Table 1 shows the baseline attributes of participants who survived and those who died. Participants who had died were more likely (*p* < 0.05) to be men and smokers. However, no significant differences in term of age, ethnicity, marital status, alcohol intake, years of education, falls history, BMI, depressive symptom, MCI and multimorbidity status was reported between participants who survived and those who died. The nutrient intake profiles between these two groups were almost similar, except that participants who died had significantly (*p* < 0.05) higher manganese intake and lower total dietary fiber and molybdenum intake than the survivals, as shown in Table 2.

Table 3 shows the levels of various trace elements in toenail and hair samples between the participants who survived and those who died. Participants who died had significantly lower zinc levels (*p* < 0.05) in toenail samples, lower calcium (*p* < 0.01) and higher lead (*p* < 0.05) levels in hair samples during baseline (Table 3). Then, we further determined the levels of biomarkers associated with oxidative stress and DNA damage among the participants to examine their association with mortality. Although the participants who died had lower SOD activities, lower MDA levels and higher DNA damage events (percentage of DNA in tail and tail moment scores) as compared to the survivals, no significant differences (*p* > 0.05) in oxidative stress (SOD activities and MDA levels) and DNA damage biomarkers (percentage of DNA in tail and tail moment scores) were detected between these two groups, as demonstrated in Table 4.

Then, by using the multivariate Cox proportional hazard analysis, lower total dietary fiber intake (HR: 0.681; 0.532–0.871), lower calcium (HR: 0.999; 95% CI: 0.999–1.000) and higher lead (HR: 1.309; 95% CI: 1.061–1.616) levels in hair samples were identified as the risk factors associated with mortality among the community-dwelling older adults in Malaysia. However, lower zinc levels in toenail samples, higher manganese and lower molybdenum intake did not appear as the predictors of mortality in this present study (*p* > 0.05), as presented in Table 5.

## 4. Discussion

This analysis aims to determine the association of trace elements’ status, oxidative stress and DNA damage biomarkers with the 5-year mortality rate among community-dwelling older adults. Our current findings showed that lower calcium and higher lead levels were associated with increased mortality among older adults. To the best of our knowledge, this is the first study demonstrating an association between calcium and lead status and the incidence of mortality among community-dwelling older adults in a multiethnic community, such as Malaysia.

The 5-year cumulative incidence of mortality among older adults aged 60 years and above was 14.4% for the overall LRGS-TUA cohort and 11.2% in this study sub-cohort. Meanwhile, the observed mortality rate for the LRGS-TUA cohort was 28.8 per 1000 person-years, which is slightly lower than the figures reported in the Statistics on Causes of Death, Malaysia [27]. Based on the report, the mortality rate of Malaysian aged 60 years and over in the years 2017, 2018, 2019 and 2020 was 36.1, 35.5, 34.8 and 32.3 per 1000 persons, respectively. As a comparison, the mortality rate among older adults aged 65 years and above in Singapore, a nation that shares some geographical, racial and cultural similarities with Malaysia, was 26.7 per 1000 persons in the year 2020 [28]. It is noted that the mortality rate of older Singaporeans should be lower if individuals aged 60–64 years old are taken into consideration. The differences in the mortality rate between Malaysians and Singaporeans could be attributed to the various discrepancies such as the socioeconomic status, the availability of healthcare facilities/infrastructures, government policies and the living environment.

Calcium is one of the essential nutrients for human health, with over 99% of calcium found in bone and teeth, while the remaining can be found in extracellular fluid and cellular organelles [29]. Calcium has many physiological roles in the body, including maintaining the normal vascular function, muscle contraction, nerve transmission, intracellular signaling, cardiac contractility, bone formation and hormonal secretion [29]. Our current findings demonstrated that lower calcium levels were associated with higher risk of mortality among community-dwelling older adults. Previously, several cohort studies and meta-analyses have demonstrated that dietary calcium intake was inversely associated with the risk of all-cause mortality [30,31,32]. Meanwhile, lower serum calcium levels (<7.9 mg/dL) during admission were associated with higher risk of in-hospital mortality among the hospitalized patients [33]. In addition, low admission serum calcium levels were associated with higher risk of all-cause long-term mortality among people with acute myocardial infarction [34]. However, it should be noted that serum calcium levels are tightly regulated under homeostasis and usually fall within the narrow physiological range [29]. Hence, serum calcium levels might not reflect the actual calcium status of the body except in the cases of severe deficiency or abnormalities in calcium homeostasis. The exact mechanisms of how lower calcium levels lead to higher risk of mortality among older adults are unclear. However, we postulated that lower calcium levels might lead to various adverse health impacts such as osteoporosis and thus increase the risk of falls, injuries, disabilities and eventually mortality among older persons [35]. Hypocalcemia is well known to be associated with various clinical manifestations, such as neuromuscular irritability (e.g., paresthesia, muscle spasms, cramps, tetany, seizures, laryngospasm and bronchospasm), cognitive and neurological impairment, personality disturbances and cardiac abnormalities [36].

In this present study, we observed that participants who died had higher lead levels than the survivors. Higher lead levels were also identified as one of the predictors of mortality among older adults. Lead is one of the common environmental toxicants in developing countries due to its persistent use in various industries and applications, such as the production and manufacturing of batteries, paints, fertilizers, cable sheaths, etc. [37]. Higher lead levels among the participants who had passed away could be due to the higher proportion of smokers in the group. Cigarette/tobacco smoke has been demonstrated as one of the routes of environmental exposure to heavy metals, including lead since the heavy metals from the soil are usually absorbed and accumulated in the plant tissues of tobacco [38]. Indeed, we found that the lead levels in smokers’ hair samples were significantly higher than in non-smokers (4.94 vs. 4.23 μg/g; *p* < 0.05). Other than smoking, inhalation of contaminated dirt particles, intake of contaminated food and water and occupational exposure are other common routes of exposure to lead toxicity [37,39].

Lead is a highly toxic metal that may cause toxicity in numerous biological systems of the body, including the renal, nervous, hematopoietic, cardiovascular, gastrointestinal and reproductive systems [37,39]. The typical clinical manifestations of lead toxicity include but are not limited to anemia, hypertension, renal dysfunction, abdominal colic, infertility, motor neuropathies, headache and neuropsychiatric impairments such as decrease in cognitive performance, loss of memory, attention deficit, confusion, depression and anxiety [37,39]. Lead exposure was also associated with various adverse events in older adults, such as age-related hearing loss, olfactory impairment, frailty, disability and functional dependence [40,41,42,43]. Recent studies have revealed the potential carcinogenic role of lead in humans [44]. In addition, lead exposure was associated with telomere length shortening and instability, which may cause cell senescence and genomic instability [45,46,47]. Previous studies have shown that individuals with shorter telomere length are at higher risk of cancer and cardiovascular disease and have a shorter life expectancy and higher mortality [48]. Indeed, studies focusing on the association between lead exposure and mortality have demonstrated that individuals with higher blood lead levels have an increased risk of all-cause, cardiovascular and cancer mortality [49,50].

This study demonstrated that total dietary fiber intake is associated with increased mortality risk among older adults. A higher intake of dietary fiber might be beneficial in promoting weight loss, reducing cholesterol levels, lowering blood pressure, improving glycemic control and inducing anti-inflammatory effects, hence improving the health condition of the older adults [51]. Dietary fiber intake is also associated with the absorption and availability of minerals and trace elements, such as calcium. Although some studies demonstrated that a higher intake of dietary fiber might decrease calcium absorption in the body, it has also been demonstrated that the fermentation of dietary fiber by bacteria residing in the lower gastrointestinal tract can facilitate the absorption of calcium [52,53]. The short-chain fatty acids produced due to the fermentation process may help to solubilize the calcium and improve the absorption of dietary calcium [53]. Moreover, previous study has demonstrated that dietary fiber intake is inversely associated with the serum heavy metal concentrations, including lead, among adults consuming recommended amounts of seafood [54]. It is postulated that dietary fiber may bind heavy metals and prevent their absorption into the body. Thus, in addition to its beneficial effects on human health, dietary fiber may affect individuals’ health and risk of mortality by modulating the absorption and availability of essential trace elements and toxic heavy metals in the human body.

The participants who died were reported to have higher manganese and lower molybdenum intake than the survivors. However, both of them did not appear as the predictors of mortality in the final multivariate Cox proportional hazard statistical model. Manganese is essential in maintaining various biological functions, including immune function, glycemic control, digestion, bone growth, blood coagulation and defense against reactive oxygen species [55]. Recently, excessive intake or exposure to manganese was reported to induce neurotoxicity and was associated with the pathogenesis of certain neurological diseases [56]. On the other hand, molybdenum is required for the proper function of the molybdenum-containing enzymes [57]. However, molybdenum deficiency is very rare, and the information on molybdenum deficiency in human health is limited.

This study has some limitations. First, the relatively smaller sample size and shorter follow-up duration of this analysis resulted in fewer recorded mortality cases. Hence, we could not perform a sub-analysis based on the specific causes of death due to the limited number of mortality cases. The smaller sample size included in this analysis means that the generalizability of the current findings may also be not possible. Hence, a future large-scale longitudinal study with a more extended follow-up period among older adults is needed to validate our current findings. Despite all these limitations, this is the first study reporting on the impact of various trace elements status on all-cause mortality in Malaysian older adults.

## 5. Conclusions

In conclusion, higher lead and lower calcium levels are associated with a higher mortality rate among community-dwelling older adults in Malaysia. Promotion of adequate calcium intake and prevention of lead exposure throughout life should be emphasized as one of the strategies for healthy aging and longevity.

## Figures and Tables

**Table 1 ijerph-19-06955-t001:** The baseline attributes of participants, total and by mortality status at 5-year follow-up.

Characteristics	Total n = 303 (100)	Survival n = 269 (88.8)	Death n = 34 (11.2)	*p*-Value
**Age**	66.66 ± 5.21	66.45 ± 5.07	68.29 ± 5.99	0.051
**Sex**				
Male	171 (56.44)	144 (53.53)	27 (79.41)	0.004 **
Female	132 (43.56)	125 (46.47)	7 (20.59)	
**Ethnicity**				
Malay	194 (64.03)	167 (62.08)	27 (79.41)	0.135
Chinese	88 (29.04)	82 (30.48)	6 (17.65)	
Indian	21 (6.93)	20 (7.44)	1 (2.94)	
**Marital status**				
Single/widowed/divorced	70 (23.10)	61 (22.68)	9 (26.47)	0.621
Married	233 (76.90)	208 (77.32)	25 (73.53)	
**Staying**				
Alone	29 (9.57)	28 (10.41)	1 (2.94)	0.163
With others	274 (90.43)	241 (89.59)	33 (97.06)	
**Smoking**				
Yes	66 (21.78)	52 (19.33)	14 (41.18)	0.004 **
No	237 (78.22)	217 (80.67)	20 (58.82)	
**Alcohol drinking**				
Yes	13 (4.29)	13 (4.83)	0 (0.00)	0.190
No	290 (95.71)	256 (95.17)	34 (100.00)	
**Education (years)**	6.81 ± 4.28	6.81 ± 4.35	6.74 ± 3.79	0.920
**Falls history**				
Yes	69 (22.77)	60 (22.30)	9 (26.47)	0.585
No	234 (77.23)	209 (77.70)	25 (73.53)	
**BMI (kg/m^2^)**	25.40 ± 4.17	25.38 ± 4.13	25.49 ± 4.56	0.895
**Depression**	2.14 ± 1.76	2.14 ± 1.76	2.15 ± 1.81	0.976
**MCI**				
Yes	31 (10.23)	26 (9.67)	5 (14.71)	0.361
No	272 (89.77)	243 (90.33)	29 (85.29)	
**Multimorbidity**				
Yes	158 (52.15)	140 (52.04)	18 (52.94)	0.921
No	145 (47.85)	129 (47.96)	16 (47.06)	

Note: Data were presented as mean ± SD or n (%). ** *p* < 0.01. BMI: body mass index; MCI: mild cognitive impairment.

**Table 2 ijerph-19-06955-t002:** The nutrient intake profiles of participants during baseline, total and by mortality status at 5-year follow-up.

Nutrient	Total n = 303 (100)	Survival n = 269 (88.8)	Death n = 34 (11.2)	*p*-Value
Energy (kcal/day)	1683.70 ± 457.66	1684.97 ± 458.25	1673.99 ± 459.85	0.896
Protein (g/day)	71.89 ± 22.18	70.92 ± 22.45	73.25 ± 20.18	0.565
Carbohydrate (g/day)	230.78 ± 72.96	230.74 ± 72.81	231.04 ± 75.18	0.982
Fat (g/day)	52.38 ± 17.96	52.61 ± 18.08	50.66 ± 17.17	0.553
Dietary fiber (g/day)	4.04 ± 2.34	4.18 ± 2.39	2.94 ± 1.57	<0.001 ***
Vitamin A (µg/day)	1358.03 ± 976.93	1356.38 ± 995.23	1370.66 ± 836.53	0.936
Vitamin C (mg/day)	122.22 ± 77.28	124.32 ± 78.80	106.19 ± 63.21	0.199
Vitamin D (mg/day)	0.27 ± 0.94	0.27 ± 0.95	0.24 ± 0.87	0.863
Vitamin E (mg/day)	6.36 ± 25.44	6.52 ± 27.04	5.13 ± 2.47	0.766
α-tocopherol (mg/day)	0.53 ± 1.32	0.56 ± 1.37	0.33 ± 0.81	0.335
Thiamin (mg/day)	1.68 ± 3.72	1.73 ± 3.86	1.26 ± 2.36	0.492
Riboflavin (mg/day)	1.26 ± 0.49	1.27 ± 0.50	1.18 ± 0.44	0.319
Niacin (mg/day)	10.77 ± 4.18	10.79 ± 4.21	10.66 ± 3.98	0.869
Pyridoxine (mg/day)	0.72 ± 0.36	0.72 ± 0.36	0.72 ± 0.34	0.948
Folate (µg/day)	109.51 ± 73.30	110.30 ± 75.20	103.53 ± 57.25	0.614
Cobalamin (µg/day)	3.61 ± 3.55	3.49 ± 3.45	4.45 ± 4.24	0.139
Pantothenic Acid (mg/day)	0.27 ± 0.49	0.28 ± 0.51	0.19 ± 0.29	0.304
Vitamin K (µg/day)	15.48 ± 48.36	15.92 ± 50.11	12.20 ± 32.36	0.674
Sodium (mg/day)	1476.84 ± 984.11	1480.98 ± 979.73	1445.20 ± 1031.59	0.842
Potassium (mg/day)	1547.69 ± 557.94	1553.18 ± 563.96	1505.71 ± 515.47	0.642
Calcium (mg/day)	530.10 ± 247.16	534.69 ± 254.28	494.99 ± 182.98	0.379
Iron (mg/day)	14.06 ± 5.66	13.99 ± 5.62	14.61 ± 6.08	0.550
Phosphorus (mg/day)	1110.57 ± 426.15	1107.73 ± 425.46	1132.31 ± 437.16	0.752
Magnesium (mg/day)	133.43 ± 66.67	133.73 ± 67.01	131.17 ± 64.95	0.834
Zinc (mg/day)	3.82 ± 1.99	3.86 ± 2.00	3.55 ± 1.93	0.399
Selenium (µg/day)	24.89 ± 18.15	25.21 ± 17.37	22.49 ± 23.47	0.412
Copper (mg/day)	0.61 ± 0.34	0.62 ± 0.35	0.55 ± 0.29	0.291
Manganese (mg/day)	0.46 ± 0.55	0.43 ± 0.55	0.64 ± 0.57	0.038 *
Molybdenum (µg/day)	0.51 ± 2.03	0.56 ± 2.14	0.14 ± 0.49	0.009 **

Note: Data were presented as mean ± SD. * *p* < 0.05; ** *p* < 0.01; *** *p* < 0.001.

**Table 3 ijerph-19-06955-t003:** Levels of various trace elements in toenail and hair samples between the participants who survived and those who died.

Characteristics.	Toenail Samples				Hair Samples			
	Total n = 303 (100)	Survival n = 269 (88.8)	Death n = 34 (11.2)	*p*-Value	Total n = 303 (100)	Survival n = 269 (88.8)	Death n = 34 (11.2)	*p*-Value
Aluminum (μg/g)	431.92 ± 242.37	434.25 ± 239.57	414.63 ± 266.21	0.689	444.73 ± 260.15	444.34 ±260.42	448.10 ± 262.47	0.942
Calcium (μg/g)	1737.82 ± 785.86	1735.03 ± 787.61	1758.48 ± 786.75	0.883	3300.72 ± 700.46	3341.45 ± 700.15	2944.36 ± 604.96	0.004 **
Cadmium (μg/g)	1.04 ± 3.44	0.94 ± 2.67	1.74 ± 6.91	0.549	0.15 ± 0.08	0.15 ± 0.08	0.14 ± 0.06	0.388
Cobalt (μg/g)	5.78 ± 11.58	5.91 ± 11.85	4.75 ± 9.51	0.618	3.86 ± 2.64	3.80 ± 2.46	4.44 ± 3.88	0.398
Iron (μg/g)	346.20 ± 246.33	348.27 ± 243.02	330.93 ± 273.87	0.727	706.60 ± 182.05	711.26 ± 181.25	665.45 ± 187.28	0.208
Lead (μg/g)	4.70 ± 2.58	4.71 ± 2.66	4.62 ± 1.93	0.869	4.43 ± 1.99	4.34 ± 1.96	5.17 ± 2.13	0.036 *
Zinc (μg/g)	94.35 ± 34.48	95.94 ± 35.46	82.59 ± 23.46	0.012 *	481.75 ± 305.57	486.33 ± 312.14	441.32 ± 240.73	0.461
Selenium (μg/g)	0.41 ± 0.18	0.41 ± 0.18	0.37 ± 0.18	0.243	0.32 ± 0.16	0.32 ± 0.16	0.32 ± 0.15	0.901
Copper (μg/g)	7.20 ± 4.52	7.25 ± 4.60	6.78 ± 3.95	0.606	12.11 ± 4.15	12.12 ± 4.13	12.05 ± 4.36	0.935
Chromium (μg/g)	42.05 ± 29.81	41.99 ± 29.52	42.47 ± 32.45	0.937	99.81 ± 32.43	99.85 ± 32.59	99.40 ± 31.55	0.945

Note: Data were presented as mean ± SD. * *p* < 0.05; ** *p* < 0.01.

**Table 4 ijerph-19-06955-t004:** Levels of biomarkers associated with oxidative stress and DNA damage between the participants who survived and those who died.

Biomarkers	Total n = 303 (100)	Survival n = 269 (88.8)	Death n = 34 (11.2)	*p*-Value
SOD	8.37 ± 6.13	8.49 ± 6.06	7.50 ± 6.73	0.378
MDA	1.98 ± 0.73	1.99 ± 0.73	1.92 ± 0.79	0.623
Percentage of DNA in Tail	12.92 ± 4.84	12.78 ± 4.81	14.00 ± 4.98	0.165
Tail moment	1.60 ± 0.81	1.58 ± 0.80	1.79 ± 0.88	0.142

Note: Data were presented as mean ± SD. MDA: malondialdehyde; SOD: superoxide dismutase.

**Table 5 ijerph-19-06955-t005:** Predictors of mortality at 5-year follow-up.

Predictors Category	Item	*p*-Value	Exp (B)	95% CI
Nutrition intake	Dietary fiber	0.002 **	0.681	0.532–0.871
Trace elements (hair samples)	Calcium	0.001 **	0.999	0.999–1.000
	Lead	0.012 *	1.309	1.061–1.616

Note: Multivariate Cox proportional hazard analysis (adjusted for age, sex, ethnicity, smoking habit and multimorbidity status). * *p* < 0.05; ** *p* < 0.01.

## Data Availability

The data presented in this study are available on request from the corresponding author.

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
