# Peer review of "Higher Lead and Lower Calcium Levels Are Associated with Increased Risk of Mortality in Malaysian Older Population: Findings from the LRGS-TUA Longitudinal Study"

_ijerph, 2022, doi:10.3390/ijerph19126955_

Round 1

Reviewer 1 Report

The manuscript “Higher Lead and Lower Calcium Levels are Associated with Increased Risk of Mortality in Malaysian Older Population: Findings From the LRGS-TUA Longitudinal Study” brings useful information about the association of various trace elements status with 5-years mortality rate among community-dwelling older adults in Malaysia. The manuscript is well written and well organized. The results are interesting and properly articulated, and the authors showed that higher lead and lower calcium and zinc levels are associated with a higher risk of mortality among community-dwelling older adults in Malaysia. The reviewer believed that the present study is interesting and potentially could contribute to the research field, however, there are only two concerns in the methods and materials, and in the conclusion. 
1. It would be better if the authors could explain and discuss methods in more detail. 
2. Major limitations of the present study were not mentioned in the conclusion section.

Author Response

Response to Reviewer 1 Comments

The manuscript “Higher Lead and Lower Calcium Levels are Associated with Increased Risk of Mortality in Malaysian Older Population: Findings From the LRGS-TUA Longitudinal Study” brings useful information about the association of various trace elements status with 5-years mortality rate among community-dwelling older adults in Malaysia. The manuscript is well written and well
organized. The results are interesting and properly articulated, and the authors showed that higher lead and lower calcium and zinc levels are associated with a higher risk of mortality among community-dwelling older adults in Malaysia. The reviewer believed that the present study is interesting and potentially could contribute to the research field, however, there are only two concerns in the methods and materials, and in the conclusion.

Point 1: It would be better if the authors could explain and discuss methods in more detail.

Response 1: We have explained and discussed the methods part in more detail as suggested by the reviewer. Please refer to the methods section (Page 2 – 4, line 83 – 181).

Point 2: Major limitations of the present study were not mentioned in the conclusion section.

Response 2: Thank you for the suggestion. However, we have mentioned the limitations of the present study in detail in the last chapter of the discussion (please refer to Page 8, line 300 – 308). We did not include the limitations of the study in the conclusion section to avoid unnecessary repetition.

Reviewer 2 Report

This elegant study focuses on the association of various trace elements status with 5-years mortality rate among community-dwelling older adults in Malaysia. The topic is relevant as pollution and waste management pose a world-wide threat to population health, and its emerging results place adequate calcium intake and prevention of lead exposure throughout life among active and healthy ageing strategies.

The findings contribute to complete and strengthen current knowledge on the impact of trace elements on the well-being and life expectancy of older populations.

The study design is linear and well grounded on previous results achieved by authors, supported by coherent references.

The major limitation is represented by the small sample size, but it is addressed in the discussion, and a future large-scale longitudinal study with a more extended follow-up period among older adults is foreseen.

The introduction (line 43) refers to age and sex as well-known unmodifiable risk factors for mortality among older populations. Genetic background should also be cited, as well as the impact of environmental factors on genes.

The results section should be improved, better describing the results supported by each table to facilitate understanding, and complementing the legends to tables/figures.

Author Response

Response to Reviewer 2 Comments

Point 1: This elegant study focuses on the association of various trace elements status with 5-years mortality rate among community-dwelling older adults in Malaysia. The topic is relevant as pollution and waste management pose a world-wide threat to population health, and its emerging results place adequate calcium intake and prevention of lead exposure throughout life among active and healthy ageing strategies.

Response 1: Thank you for the comments.

Point 2: The findings contribute to complete and strengthen current knowledge on the impact of trace elements on the well-being and life expectancy of older populations.

Response 2: Thank you for the comments.

Point 3: The study design is linear and well grounded on previous results achieved by authors, supported by coherent references.

Response 3: Thank you for the comments.

Point 4: The major limitation is represented by the small sample size, but it is addressed in the discussion, and a future large-scale longitudinal study with a more extended follow-up period among older adults is foreseen.

Response 4: Thank you for the comments.

Point 5: The introduction (line 43) refers to age and sex as well-known unmodifiable risk factors for mortality among older populations. Genetic background should also be cited, as well as the impact of environmental factors on genes.

Response 5: We have acknowledged the impact of genetic factors and interaction of these genes with environmental factors on human longevity in the introduction part as suggested by the reviewer. Please refer to Page 2, line 48 – 50.

Point 6: The results section should be improved, better describing the results supported by each table to facilitate understanding, and complementing the legends to tables/figures.

Response 6: Thank you for the suggestion. We have rewritten the results section to better describe the findings of the study. Please refer to Page 4 – 5, line 182 – 208.
